# BEYOND FIXED BUDGETS: DYNAMIC REASONING EFFICIENCY REWARD FOR LARGE LANGUAGE MODEL

## ABSTRACT

The "slow thinking" paradigm has been widely validated to enhance the reasoning capabilities of large language models, but it also introduces reasoning inefficiency: models may overthink simple problems while prematurely shifting their reasoning paths when tackling complex problems. To address this, we propose AdapThink, a simple yet efficient post-training framework designed to control preferences for "slow thinking" pattern adaptively. Unlike directly imposing length budgets or setting overlong filters, AdapThink leverages group-level length distributions and reflective word distributions to construct reasoning process rewards and introduce a two-stage sampling strategy aimed at maximizing group diversity. Experimental results demonstrate that when post-training two DeepSeek-distilled Qwen models under a context length limit of only 2K tokens, AdapThink achieves a 27% improvement in convergence rewards compared to the GRPO baseline. Notably, when testing models with a 32K token limit, AdapThink also achieves 12.6% improvement over base model in several mathematical benchmarks.

## 1 INTRODUCTION

The breakthroughs in large language models (LLMs) such as OpenAI's o1 (OpenAI, 2024) and DeepSeek R1 (Guo et al., 2025) have demonstrated that reinforcement learning (RL)-based post-training methods can substantially improve their reasoning capabilities. This enhancement is primarily attributed to the emergence of sophisticated self-reflection behaviors in models (Kumar et al., 2025; Kazemnejad et al., 2024). However, recent research has highlighted a significant inefficiency associated with this "*slow thinking*" pattern (Muennighoff et al., 2025; Han et al., 2024; Chen et al., 2024; Wang et al., 2025b; Aggarwal & Welleck, 2025; Shen et al., 2025b). Reasoning models frequently *overthink* simple problems, spending unnecessary computational resources, while conversely *underthinking* complex challenges, leading to incomplete reasoning and incorrect answers. When presented with a simple problem (see Appendix A), a model can reach a correct answer with merely 479 tokens. Yet, its self-validation mechanisms—marked by phrases such as "Verify" and "Wait"—triggered unnecessary reflections, resulting in more than a quadrupled token consumption. In contrast, when attempting a complex problem, the same model exhibited frequent and unproductive shifts, marked by phrases "Alternatively" and "Another." Upon reaching its token limit, the model stopped and arrived at an incorrect answer.

Therefore, an ideal chain of thought (CoT) would be capable of adjusting its self-reflection frequency and depth, adaptive to the problem difficulty and its level of confidence. Empirically, (Ma et al., 2025; Xie et al., 2025) have demonstrated that employing reflection vocabularies does *not always* guarantee mathematical reasoning improvement. Drawing upon this insight, recent works explored direct control via modifying the input prompt (Jin et al., 2024; Muennighoff et al., 2025; Liu et al., 2025; Han et al., 2024; Chen et al., 2024) or providing indirect reasoning-length rewards (Wang et al., 2025b; Aggarwal & Welleck, 2025; Shen et al., 2025b). A common limitation of these approaches is using token budgets governed by rules or offering rewards for adhering to a special *length budget*, overlooking the critical impact of the changes in models' capabilities and length preferences. Intuitively, models with limited reasoning abilities benefit from more extended CoT patterns, as redundant self-reflection could serendipitously contribute to reaching correct solutions. In contrast, high-performing models should aim to minimize token consumption to prevent overthinking and to maintain efficient CoT reasoning.

Motivated by these insights and our observations in Section 2, we propose AdapThink, a token-efficient post-training framework for reasoning models. Our post-training framework allows models to tailor their reasoning depth preference to align with their current operational capabilities. Instead of directly limiting the budget of reasoning length, our work adjusts the length and reflection preference through analyzing the distribution of diverse reasoning patterns observed in groups of generated samples. We reveal that **control reasoning length alone does not directly determine model reasoning efficiency.** Instead, strategically regulating reasoning depth is a more effective way to mitigate overthinking and underthinking. Overall, our main contributions are:

(1) First, we introduce a group-level reasoning process reward. Instead of setting a fixed token budget or restricting the number of reflection words, we quantify reflection-related preferences by analyzing statistical differences in response length and reflection word usage within groups.

(2) Furthermore, we propose a diversity sampling mechanism to accelerate learning efficiency and enhance response diversity. Unlike existing oversampling strategies designed for answer-based rewards, our AdapThink considers the entropy scores of reasoning process-based rewards, aiming to increase CoT diversity while satisfying group accuracy constraints.

(3) By post-training two DeepSeek-Distilled Qwen models with a context length limit of only *2K tokens*, our method outperforms multiple length-control baselines under a *32K-token* limit. Experimental analyses demonstrate that AdapThink learns to generate efficient CoT rather than merely shortening text to meet fixed length restrictions.

## 2 OBSERVATIONS

To investigate potential overthinking and underthinking issues in current reasoning language models, we first conducted a comprehensive analysis of the generation patterns of the DeepSeek-R1-Distill-Qwen-1.5B and DeepSeek-R1-Distill-Qwen-7B models (Guo et al., 2025) on the MATH-500 mathematical dataset. We measure the efficiency of CoT from two perspectives:

**DeepSeek-R1-Distill-Qwen-1.5B**

| wait | but | alternatively | let me think | hmm |
|------|------|------|------|------|
| 14039 | 13476 | 2515 | 1760 | 1623 |
| another | check | hold on | make sure | actually |
| 1310 | 1198 | 907 | 769 | 726 |
| verify | let me see | why | instead | confirm |
| 297 | 235 | 127 | 108 | 70 |
| ensure | however | evaluate | examine | yet |
| 39 | 37 | 30 | 24 | 23 |
| while | rather | on the other hand | though | otherwise |
| 22 | 13 | 10 | 7 | 6 |

**DeepSeek-R1-Distill-Qwen-7B**

| but | wait | hmm | alternatively | another |
|------|------|------|------|------|
| 6938 | 6191 | 1609 | 1514 | 1168 |
| check | let me think | make sure | actually | hold on |
| 1072 | 674 | 672 | 605 | 532 |
| verify | let me see | confirm | why | instead |
| 368 | 255 | 148 | 113 | 106 |
| ensure | however | while | evaluate | otherwise |
| 71 | 65 | 45 | 41 | 25 |
| yet | though | examine | rather | on the other hand |
| 21 | 21 | 16 | 9 | 7 |

Figure 1: **The Reflection-related Words Distribution** of the responses generated by the DeepSeek-Distill-Qwen 1.5B (left) and 7B (right) models on the MATH-500 dataset, with an 8K token limit and 8 rollouts per question. Each cell represents a specific reflection-related word and its average occurrence frequency in the responses.

**Token Length Distribution.** Following the settings in (Aggarwal & Welleck, 2025), we set the maximum token limit to 8,192 during inference and divide the response range into four equal intervals to analyze the distribution patterns. We then conduct statistical analysis on the samples with correct and incorrect answers separately across different bin counts.

**Reflection Words Distribution.** The occurrence count of reflection-related terms is commonly used to verify the emergence of reasoning capabilities in models (Guo et al., 2025). Studies such as (Ma et al., 2025; Xie et al., 2025; Aggarwal & Welleck, 2025) all adopt specific reflection-related terms as quantitative metrics for reasoning effectiveness. Therefore, we summarized and presented the occurrence counts of most reflection-related terms in the responses. We selected the top 25 reflection-related words from the responses and analyzed their distribution, as shown in Figure 1. Similarly, we divided the number of reflection words appearing in each response into four equal intervals and conducted statistical analysis by distinguishing between correct and incorrect responses.

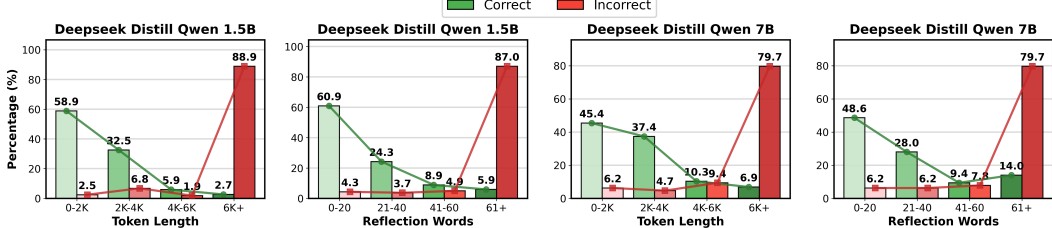

Figure 2: **Key Observations** from the distribution analysis of the DeepSeek-Distill-Qwen 1.5B and 7B models on the MATH-500 dataset. From left to right: token length ranges and reflection word frequency intervals for each model. The y-axis represents the percentage distribution of correct vs. incorrect samples.

As shown in Figure 2, **over 83% of incorrect answers** correlate with higher token consumption and increased usage of reflection-related words. This suggests that when confronting challenging problems, models tend to exhibit uncertain behavior through alternative explorations and repetitive self-verifications triggered by reflection-related terms. Meanwhile, for correct answers, we observed that **approximately 54% of correct responses** predominantly fall within the lowest token length bin and show lower frequencies of reflection word occurrences, suggesting a potential distinction between the preference for reflection words in correct and incorrect responses.

These observations motivate us to explore the value of the emergence of reflection-related words for reasoning capabilities and reasoning efficiency. Instead of focusing on the token level, we conduct our research at the sequence level, aiming to adjust the preference for reflection based on the overall statistical information of reflection-related terms within a response.

## 3 METHOD

Building on the observations and analyses presented in Section 2, this section elaborates on our proposed post-training framework AdapThink, focusing on addressing two critical questions:

**Q1.** How to measure the degree of reflection without relying on fixed-length budgets? And how to allocate higher rewards to more efficient CoT beyond correctness-based rewards?

**Q2.** How to enhance the potential diversity of reasoning patterns for each question, and how to better leverage reasoning diversity to accelerate the learning of efficient CoT?

We address Q1 and Q2 in Sections 3.1 and 3.2, respectively, with the overall framework of Adap-Think illustrated in Figure 3 and the pseudo-code provided in Appendix B.

### 3.1 GROUP-RELATIVE REASONING PROCESS REWARD

We begin with a pre-trained reasoning language model $\pi_\theta$ and a dataset $\mathcal{D} = \{(x^k, y^{*k})\}_{k=1}^N$, where each instance contains the input prompt $x$ and the correct answer $y^*$. For each input $x \in \mathcal{D}$, $\pi_\theta$ performs reasoning to generate $|\mathcal{G}|$ samples $\mathcal{G} := \{\hat{y}_i\}_{i=1}^{|\mathcal{G}|}$.

For each generated sample $\hat{y}_i \in \mathcal{G}$, we define two key metrics to characterize its reasoning pattern:

- $l(\hat{y}_i)$: the total number of tokens in the reasoning chain of $\hat{y}_i$

Figure 3: **The Framework Architecture of AdapThink**. The left semicircular shapes ◖ represent "prompts", and the right semicircular shapes ◗ represent "responses". Green ◗ indicates correct answers, while red ◗ indicates incorrect answers. Different shades correspond to varying reasoning process rewards based on Equation 4.

- $r(\hat{y}_i)$: the count of reflection-related words in $\hat{y}_i$, including terms such as "wait", "alternatively", "verify", and other metacognitive expressions (see Figure 5)

Next, based on the metrics $l(\hat{y}_i)$ and $r(\hat{y}_i)$ of each response, we analyze the output correctness and CoT characteristics of each group $\{\hat{y}_i\}_{i=1}^{|\mathcal{G}|}$.

**Outcome Correctness.** We represent the average correctness of the model's $|\mathcal{G}|$ samples within a group as the response accuracy, denoted as $\varphi$:

$$\varphi(x, \hat{y}_{1,\ldots,|\mathcal{G}|}) = \frac{1}{|\mathcal{G}|} \sum_{i=1}^{|\mathcal{G}|} \mathbb{I}(\hat{y}_i = y^*), \tag{1}$$

where $\mathbb{I}(\cdot)$ is the indicator function and $y^*$ is the ground truth answer.

**Process Characteristics.** Given the distinct reasoning patterns observed between correct and incorrect answers in Section 2, we partition the response groups into correct ($\mathcal{G}_\mathcal{T}$) and incorrect ($\mathcal{G}_\mathcal{F}$) sub-groups based on their correctness:

$$\mathcal{G}_\mathcal{T} = \{\hat{y}_i \in \mathcal{G} : \hat{y}_i = y^*\}, \quad \mathcal{G}_\mathcal{F} = \{\hat{y}_i \in \mathcal{G} : \hat{y}_i \neq y^*\} \tag{2}$$

For each sub-group, we compute statistical measures of key reasoning process metrics:

$$\mu_l(\mathcal{G}) = \frac{1}{|\mathcal{G}|} \sum_{\hat{y}_i \in \mathcal{G}} l(\hat{y}_i), \quad \mu_r(\mathcal{G}) = \frac{1}{|\mathcal{G}|} \sum_{\hat{y}_i \in \mathcal{G}} r(\hat{y}_i) \tag{3}$$

where $\mathcal{G} \in \{\mathcal{G}_\mathcal{T}, \mathcal{G}_\mathcal{F}\}$. Thus, $\mu_l(\mathcal{G})$ and $\mu_r(\mathcal{G})$ represent the mean reasoning lengths and reflection word frequencies for each correctness group, respectively.

Additionally, the group correctness metric $\varphi$ reflects the model's ability to tackle the current question $x$. Intuitively, when the model demonstrates high $\varphi$, the responses $\hat{y}$ characterized by correct answer, shorter reasoning length, and fewer reflection-related terms typically indicate a more efficient and direct problem-solving process.

To verify our hypothesis, we introduce a group-level process reward $r_{\text{ref}}(\hat{y}_i)$ that adaptively regulates reasoning preference based on group correctness $\varphi$ and the CoT characteristics:

$$r_{\text{ref}}(\hat{y}_i) = \text{clip}\left(\omega(\varphi) \cdot \left[\left(\frac{\mu_l(\mathcal{G}(\hat{y}_i)) - l(\hat{y}_i)}{\mu_l(\mathcal{G}(\hat{y}_i))}\right) + \left(\frac{\mu_r(\mathcal{G}(\hat{y}_i)) - r(\hat{y}_i)}{\mu_r(\mathcal{G}(\hat{y}_i))}\right)\right], -1, 1\right) \tag{4}$$

where $\mathcal{G}(\hat{y}_i) \in \{\mathcal{G}_\mathcal{T}, \mathcal{G}_\mathcal{F}\}$ denotes the correctness group that sample $\hat{y}_i$ belongs to. Each fraction $\frac{\mu - \text{individual}}{\mu}$ measures the relative difference between the group mean and the individual sample's reasoning characteristics, normalized by the group mean. Positive values indicate that the sample uses fewer tokens/reflection words than the group average, while negative values indicate the opposite.

The weight function $\omega(\varphi)$ determines whether to favor more efficient CoT based on group correctness:

$$\omega(\varphi) = \begin{cases} 0 & \text{if } \varphi < \varphi_{\text{thr}} \\ \cos\left(\frac{\pi}{2} \cdot \frac{1-\varphi}{1-\varphi_{\text{thr}}}\right) & \text{if } \varphi_{\text{thr}} \leq \varphi \leq 1 \end{cases} \tag{5}$$

In Equation 5, if the current group's correctness $\varphi$ falls below the threshold $\varphi_{\text{thr}}$, we do not impose any preference on the reasoning process. This design ensures that when dealing with models lacking reasoning capabilities or tackling difficult problems, we still train them in **an answer-pursuing mode**. Once $\varphi$ increases from $\varphi_{\text{thr}}$ to 1, the weight assigned to prioritizing efficient CoT, characterized by fewer reflection-related terms and shorter response lengths, increases gradually through a smooth cosine-based transition.

## 3.2 DIVERSITY-AWARE SAMPLING

As shown in Equation 4, the reasoning process reward incorporates group-relative distribution patterns from $\mathcal{G}_{\mathcal{T}}$ and $\mathcal{G}_{\mathcal{F}}$ into its computation. Therefore, the diversity of reasoning metrics $l(\hat{y})$ and $r(\hat{y})$ within each group significantly influences the effectiveness of $r_{\text{ref}}$ learning. While previous studies have addressed the zero-advantage problem through dynamic oversampling strategies (Yu et al., 2025) to avoid homogeneous rewards within groups, they have not considered the diversity of reasoning processes. We claim that **homogeneous reasoning patterns may also hinder effective model learning by limiting the model's exposure to diverse problem-solving strategies.**

To address this limitation, we propose a two-stage sampling strategy that enhances reasoning diversity while maintaining balanced group correctness. The pseudo-code of this sampling strategy is provided in Appendix Algorithm 2.

**Stage 1: Upsampling and Diversity Measurement.** First, we oversample by a factor of $K$ to expand the candidate response pool to $\mathcal{G}'$ for current question $x$, enabling exploration of diverse reasoning styles within each group. To quantify this diversity, we introduce an entropy-based metric $\mathcal{H}$ that captures the distributional spread of reasoning length $l(\hat{y}_i)$ and reflection word count $r(\hat{y}_i)$.

Specifically, we partition the combined range of $l(\hat{y}_i)$ and $r(\hat{y}_i)$ into four bins $\mathcal{S}$ as defined in Section 2, where each bin $s \in \mathcal{S}$ corresponds to a unique interval combination of length and reflection word count. The diversity metric $H$ is then defined as:

$$\mathcal{H}(\mathcal{G}') = -\frac{1}{\log |\mathcal{G}'|} \sum_{s \in \mathcal{S}} \frac{|s|}{|\mathcal{G}'|} \log \frac{|s|}{|\mathcal{G}'|} \tag{6}$$

where $|s|$ counts the number of samples in $\mathcal{G}'$ whose $r(\hat{y}_i)$ values fall into the interval pair of bin $s$. Higher $\mathcal{H}(\mathcal{G}')$ indicates more diverse reasoning patterns in a group.

**Stage 2: Diversity-Maximizing Downsampling.** From the upsampled pool $\mathcal{G}'$, we downsample to a final set $\mathcal{G}$ of target size $|\mathcal{G}|$. Specifically, we select samples to maximize the diversity entropy $H(\mathcal{G})$ while ensuring the response accuracy of the downsampled group satisfies $\varphi(\mathcal{G}) \in (0, 1)$ if the original $\varphi(\mathcal{G}') \in (0, 1)$.

This design serves two purposes: (1) the retained samples maintain maximal diversity in reasoning length $l(\hat{y}_i)$ and reflection word count $r(\hat{y}_i)$, providing rich training signals for the reasoning process reward; (2) the constraint $\varphi(\mathcal{G}) \in (0, 1)$ avoids extreme homogeneity in answer correctness (i.e., neither all correct nor all incorrect), thus alleviating the zero-advantage problem and ensuring meaningful reward gradients.

## 4 EXPERIMENTAL SETUP

**Datasets and Base Models.** We conduct experiments on a curated lightweight mathematics dataset that spans various difficulty levels. This dataset combines queations from *DeepScaleR-Preview-Dataset* (Luo et al., 2025b), including about 5K question-answer pairs sampled from AIME (1984-2023), AMC (prior to 2023), and MATH training sets. For baseline models, we employ *DeepSeek-R1-Distill-Qwen-1.5B* and *DeepSeek-R1-Distill-Qwen-7B* (Guo et al., 2025), which are obtained through supervised fine-tuning on reasoning data generated by the DeepSeek-R1 model.

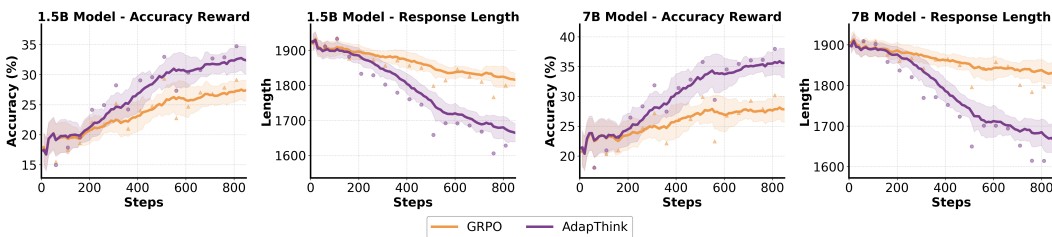

Figure 4: **Training Comparison between GRPO and AdapThink** on DeepSeek-Distill-Qwen-1.5B and DeepSeek-Distill-Qwen-7B with a 2K token response length constraint.

Table 1: **Performance Comparison on Mathematical Reasoning Benchmarks.** The **Acc** metric is determined by averaging `pass@1` scores over 16 independent generation runs, and the **Token** metric measures the corresponding average response length for each benchmark. Overall results are the simple arithmetic mean across all benchmarks.

| Method | MATH-500 | | AMC 2023 | | AIME 2024 | | AIME 2025 | | Overall | |
|---|---|---|---|---|---|---|---|---|---|---|
| | Acc↑ | Tokens↓ | Acc↑ | Tokens↓ | Acc↑ | Tokens↓ | Acc↑ | Tokens↓ | Acc↑ | Tokens↓ |
| *DeepSeek-R1-Distill-Qwen-1.5B* | | | | | | | | | | |
| Original | 83.0 | 3,978 | 67.7 | 7,160 | 29.3 | 13,832 | 26.9 | 14,680 | 51.73 | 5,348 |
| O1-Pruner (Luo et al., 2025a) | 82.3 | 2,446 | 69.5 | 5,622 | 27.5 | 12,155 | 24.1 | 12,701 | 50.85 | 3,787 |
| No Wait (Wang et al., 2025a) | 81.6 | 2,894 | 68.5 | 6,422 | 26.1 | 9,167 | 20.3 | 11,601 | 49.13 | 4,048 |
| DAST (Shen et al., 2025b) | 83.4 | 3,155 | 70.2 | 5,583 | 29.5 | 10,042 | 20.6 | 10,647 | 50.93 | 4,139 |
| FCS+Ref. (Chen et al., 2024) | 82.9 | 2,883 | 72.1 | 4,449 | 29.7 | 9,898 | 28.6 | 11,624 | 53.32 | 3,820 |
| LCPO (Aggarwal & Welleck, 2025) | 80.5 | 2,684 | 63.9 | 6,694 | 23.5 | 9,692 | 24.7 | 10,075 | 48.15 | 3,873 |
| DEER (Yang et al., 2025) | 84.1 | 2,398 | 70.2 | 4,179 | 26.6 | 9,732 | 20.2 | 9,890 | 50.27 | 3,320 |
| MERA (Ha et al., 2025) | 86.3 | 2,165 | 73.3 | 3,262 | 31.4 | 8,927 | 33.4 | 9,495 | 56.10 | 2,964 |
| **AdapThink (Ours)** | **86.8** | 2,742 | **76.0** | 4,226 | **33.3** | 9,521 | **36.0** | 9,719 | **58.03** | 3,575 |
| *DeepSeek-R1-Distill-Qwen-7B* | | | | | | | | | | |
| Original | 86.9 | 3,422 | 77.4 | 6,738 | 53.1 | 12,185 | 48.2 | 13,276 | 66.40 | 4,719 |
| O1-Pruner (Luo et al., 2025a) | 89.0 | 2,678 | 82.8 | 7,501 | 52.2 | 9,412 | 49.4 | 10,973 | 68.35 | 4,002 |
| No Wait (Wang et al., 2025a) | 87.2 | 2,579 | 75.7 | 5,478 | 42.5 | 10,048 | 35.2 | 11,827 | 60.15 | 3,733 |
| DAST (Shen et al., 2025b) | 88.6 | 2,876 | 80.3 | 4,601 | 52.6 | 10,240 | 49.5 | 9,721 | 67.75 | 3,762 |
| FCS+Ref. (Chen et al., 2024) | 87.8 | 2,909 | 81.5 | 5,143 | 55.1 | 9,212 | 49.8 | 9,844 | 68.55 | 3,815 |
| LCPO (Aggarwal & Welleck, 2025) | 84.7 | 2,539 | 73.6 | 4,821 | 45.3 | 9,408 | 40.6 | 9,904 | 61.05 | 3,498 |
| DEER (Yang et al., 2025) | 88.9 | 1,908 | 82.2 | 5,194 | 46.4 | 9,932 | 39.6 | 9,302 | 64.28 | 3,052 |
| MERA (Ha et al., 2025) | 91.0 | 1,739 | 85.7 | 3,711 | 56.1 | 8,398 | **50.6** | 8,732 | 70.85 | 2,631 |
| **AdapThink (Ours)** | **92.0** | 2,472 | **90.0** | 4,047 | **60.4** | 9,464 | 49.1 | 9,864 | **72.88** | 3,346 |

**Comparing Methods.** We conduct post-training on DeepSeek-R1-Distill-Qwen-1.5B and DeepSeek-R1-Distill-Qwen-7B using Group Relative Policy Optimization (GRPO) (Guo et al., 2025) as the base algorithm. The detailed objective is shown in Appendix B. To evaluate different length-control methods for enhancing reasoning efficiency, we implement the following baselines: (1) Methods that **directly reduce reasoning length**, including O1-Pruner (Luo et al., 2025a) and No-Wait (Wang et al., 2025a); (2) Methods that **rely on preset computational budgets** before inference, including DAST (Shen et al., 2025b), FCS+Ref (Chen et al., 2024), and LCPO (Aggarwal & Welleck, 2025); (3) Methods that **dynamically determine termination**, including DEER (Yang et al., 2025) and MERA (Ha et al., 2025).

## 5 RESULTS

In this section, we evaluate and analyze the performance of AdapThink across various settings through three key aspects: answer accuracy, reasoning efficiency, and group diversity.

**Overall Comparison.** As shown in Figure 4 and Table 1, AdapThink demonstrates superior performance across both model scales. Compared to the original baseline, AdapThink achieves a **6.11%** relative improvement on the 1.5B model and a **7.27%** relative improvement on the 7B model.

Notably, although trained under a strict 2K token limit, AdapThink develops a dynamic preference for reasoning depth based on group characteristics, rather than a fixed budget. Consequently, when

evaluated with a 32K token context, the post-trained model adaptively tailors its response length to the complexity of the problem. As shown in Table 1, for challenging benchmarks like AIME, AdapThink generates longer responses to facilitate complex reasoning, while maintaining conciseness on simpler problems such as those in MATH-500.

While MERA achieves shorter response lengths with a 17.1% reduction on the 1.5B model and 21.4% reduction on the 7B model, its performance also drops relative to ours by **1.93** and **2.03**, respectively. Particularly on the AMC 2023 and AIME 2024, 2025 benchmarks, our performance significantly outperforms MERA.

Table 2: **Test Performance Comparison of AdapThink Variants.** `Max Token` represents the maximum response token limit (in thousand) in training phase. The accuracy metric is determined by averaging `PASS@1` scores over 16 independent generation runs under the 8K token limit on AIME 2025. The diversity metrics are from the training phase, while other metrics are from testing. $\mathcal{G}_{\mathcal{T}}$ and $\mathcal{G}_{\mathcal{F}}$ indicate correct and incorrect answer groups, respectively.

| AdapThink Variants | | | | Accuracy | Length | | Reflection Words | | Diversity | |
| --- | --- | --- | --- | --- | --- | --- | --- | --- | --- | --- |
| $\mu_l$ | $\mu_r$ | $K$ | Max Token | PASS@1 | $\mu_l(\mathcal{G}_{\mathcal{T}})$ | $\mu_l(\mathcal{G}_{\mathcal{F}})$ | $\mu_r(\mathcal{G}_{\mathcal{T}})$ | $\mu_r(\mathcal{G}_{\mathcal{F}})$ | $H(\mathcal{G}_{\mathcal{T}})$ | $H(\mathcal{G}_{\mathcal{F}})$ |
| ✓ | ✓ | 2 | 2 | 25.4 ↑7.5 | 3561 ↓580 | 7442 ↓478 | 18.5 ↓3.0 | **34.5** ↓4.1 | 25.7 ↑3.2 | 32.1 ↓1.8 |
| ✗ | ✓ | 1 | 2 | 23.3 ↑5.4 | 4772 ↑630 | 8154 ↑235 | 30.3 ↑5.8 | 40.3 ↑1.7 | 22.2 ↓0.3 | 32.1 ↓1.8 |
| ✓ | ✗ | 1 | 2 | 22.9 ↑5.0 | 4395 ↓253 | 7872 ↑48 | 30.4 ↑6.5 | 40.9 ↑2.3 | 23.5 ↑1.0 | 29.3 ↓4.6 |
| ✓ | ✓ | 1 | 2 | 25.0 ↑7.1 | 3803 ↓339 | 7579 ↓340 | 22.4 ↓1.4 | 35.7 ↓2.9 | 23.2 ↑0.8 | 33.5 ↓0.4 |
| ✓ | ✓ | 2 | 4 | 24.6 ↑6.7 | 3861 ↓281 | 7909 ↑10 | 27.3 ↑3.4 | 46.1 ↑7.5 | 24.5 ↑2.1 | 27.4 ↓6.5 |
| ✓ | ✓ | 2 | 2→4 | **26.0** ↑8.1 | **3339** ↓802 | **6886** ↓1034 | **15.5** ↓8.3 | 38.6 ↓0.0 | **31.6** ↑9.1 | **30.3** ↓3.6 |

To verify the effectiveness of each component in the AdapThink framework, we conducted comprehensive ablation experiments, and the evaluation results are shown in Figure 5 and Table 2.

**Ablation of Diversity-aware Sampling $\mathcal{H}$.** As shown in Appendix Table 4 and the figure right, AdapThink (w/o $\mathcal{H}$), which is equivalent to GRPO + $r_{\text{ref}}$, exhibits slower convergence. While this has a smaller impact on the test set, as seen in Table 2, Row 4, where AdapThink (w/o $\mathcal{H}$) still achieves a **+7.1** PASS@1 accuracy improvement over the baseline. We further verified the general applicability of our proposed diversity sampling method by applying it to other length-control methods. As shown in Table 4, we found that this sampling method also **accelerates the convergence** of these baselines: LCPO+$\mathcal{H}$ exhibits a 36.4% improvement in convergence speed, and DAST+$\mathcal{H}$ shows a 31.6% improvement.

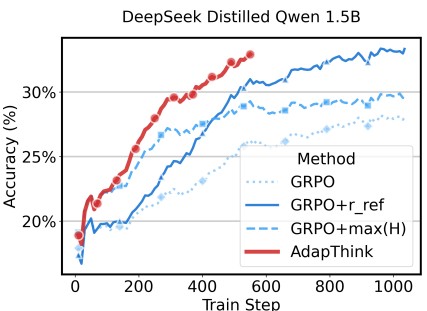

**Ablation of Reasoning Process Reward $r_{\text{ref}}$.** As shown in Equation 4, the $r_{\text{ref}}$ is controlled by metrics $\mu_l$, $\mu_r$, and weight $w(\varphi)$. We first study $\mu_l$ and $\mu_r$ through ablation, as shown in Rows 2,3 of Table 2. Using only $\mu_r$ or $\mu_l$ alone cannot fully control the reduction of reflection words, but they also bring **5.4% and 5.0% performance gains** over the original base. However, the complete AdapThink method uses both parts together, achieving a better performance gain while controlling reflection words more effectively.

For $w(\varphi)$, we conduct the ablation AdapThink(no weight) with the training dynamic shown in Figure 5 and test dynamic in Appendix Figure 7. AdapThink(no weight) **greatly reduced** response lengths, which is helpful in the early stages of training due to the deeply overthinking phenomenon in the base model. However, the later training stage of AdapThink(no weight) led to **significantly degraded performance** on the AIME 2025 benchmark. As shown in Figure 7, we found that the shorter length in later stages of AdapThink(no weight) matched with lower `PASS@1`. It uncovers a notable insight for current length-control methods: *Consistently prioritizing shorter lengths does not benefit reasoning performance; while enabling reflection-related preferences at the appropriate time can improve overall effectiveness without sacrificing model performance.*

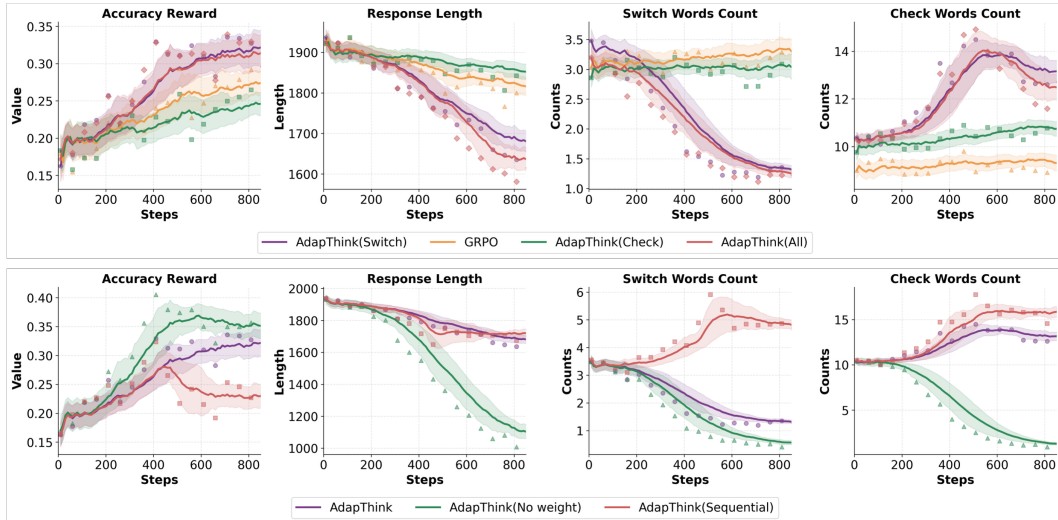

Figure 5: **Training Comparison of AdapThink Variants** with different calculation methods for $r_{\text{ref}}$ (Equation 4). AdapThink(Switch), AdapThink(Check), and AdapThink(Sequential) denote controlling different transition words or conjunctions as $\mu_r$, while AdapThink(No weight) indicates the elimination of weighting by $w(\varphi)$.

**The Potential Impact of the Selection of Reflection-related Terms on the Metric** $r(\hat{y})$. We further conduct ablations on different types of reflection-related or non-reflection-related term controls to examine their impact on reasoning patterns when calculating $r(\hat{y})$. The selection criteria follow those in (Yeo et al., 2025; Luo et al., 2025b), including AdapThink(check) for verification-related words ("*wait*", "*hold on*", "*check*", and "*verify*"), AdapThink(switch) for reasoning expansion-related transition words ("*alternatively*", "*however*", "*another*", and "*instead*"), and a reflection-unrelated ablation AdapThink(sequential) for sequential transition words without reflection meaning ("*first*", "*then*", "*next*", "*finally*", "*therefore*", "*so*", "*thus*"). As shown in Figure 5, Adap-Think(Switch) functions similarly to AdapThink. This aligns with the motivation outlined in the study by (Wang et al., 2025b): Frequent thought switching exacerbates the model's propensity to generate incorrect answers. AdapThink addresses this issue by appropriately controlling its preference for such words. Additionally, AdapThink(sequential) exhibits no clear changes in either outcome rewards or process metrics, which aligns with our expectations, as reflection-unrelated terms exert little impact on reasoning patterns. Meanwhile, controlling via AdapThink(check) actually impairs model performance, which indicates that the model's hesitation behavior is crucial for the reasoning patterns required to generate correct answers.

## 5.1 DISCUSSION

**Curriculum Learning.** Inspired by (Luo et al., 2025b), we implemented a curriculum learning for AdapThink. Specifically, we conduct secondary training on the best checkpoint from the 2K token limit to 4K. For comparison, we also trained a model directly with 4K token limit using AdapThink. Our results (Table 2 Rows 5,6) shows that progressive training from 2K to 4K token limit outperformed direct 4K training, and achieves the highest accuracy while maintaining the most efficient reasoning patterns among all the AdapThink variants.

Table 3: N-gram repetition analysis across different post-training methods.

| Method | N-grams (%) | | |
|---|---|---|---|
| | Total | $\mathcal{G}_{\mathcal{T}}$ | $\mathcal{G}_{\mathcal{F}}$ |
| GRPO | **0.3** | **0.3** | **0.2** |
| DAST | 3.0 | 3.5 | 2.2 |
| LCPO | 10.8 | 5.9 | 19.8 |
| AdapThink | 0.7 | 0.6 | 0.9 |

**Reward Hacking.** To further investigate potential reward hacking introduced by length-control mechanisms, we employ N-gram ($N = 40$) repetition rate metrics (Yeo et al., 2025) to quantify repetitive patterns in model responses for the MATH-

500 test dataset, as shown in Table 3. Our analysis reveals that LCPO exhibits severe repetitive patterns with an average N-gram repetition rate of 10.8%, especially 19.8% in incorrect answers, indicating potential reward hacking in its length control mechanism. Notably, AdapThink maintains **consistently low repetition rates** across both correct and incorrect responses, suggesting its superior robustness for scaling across more models.

**Out-of-Distribution Evaluation.** We also validated AdapThink on non-math out-of-distribution (OOD) benchmarks in Appendix Table 5. We found that AdapThink's improvements over the base model are not limited to math tasks. It also achieves performance gains on OOD tasks with improved **2.77%** Pass@1 scores and reduced **26.7%** token usage.

## 6 RELATED WORK

**Induce Longer CoT.** For inducing longer reasoning length, several works (Weng et al., 2023; Miao et al., 2023; Saunders et al., 2022; Renze & Guven, 2024b; Jin et al., 2024) have encouraged models to engage in deeper thinking through natural language feedback. For zero-shot CoT, (Weng et al., 2023; Miao et al., 2023) stimulate model self-reflection by performing backward verification or multiple response voting. For few-shot CoT with demonstrations, (Jin et al., 2024) introduced five general standardized patterns to induce models to simulate human thinking and reshape the CoT. However, these one-size-fits-all approaches ignore the diversity of problem-solving paths, only have a monotonous processing mode. Similarly, (Muennighoff et al., 2025) designed several budget forcing mechanisms to increase guidance in CoT, such as appending "Wait" multiple times when the model tries to end, forcing it to double-check. Likewise, (Shen et al., 2025a) introduced special meta-action markers like `<|continue|>`, `<|reflect|>`, and `<|explore|>`, enabling the model to restart from intermediate steps, lengthen responses, and correct errors.

**Induce Shorter CoT.** Simple prompt methods are also effective for encouraging a shorter CoT (Nayab et al., 2024; Xu et al., 2025; Renze & Guven, 2024a). For example, (Muennighoff et al., 2025) added "Final answer" to terminate the model's thinking process. Besides, (Jin et al., 2024; Kang et al., 2025) used stronger models to compress long CoTs semantically into shorter ones. (Yeo et al., 2025; Shen et al., 2025b; Aggarwal & Welleck, 2025; Yu et al., 2025) design different length-budget signals from the perspective of rule-based reward design, encouraging models to balance accuracy and token efficiency during the thinking process. Although (Shen et al., 2025b) already takes problem complexity and model confidence into account for the budget, its adoption of a uniformly shorter response preference is less suitable for models with weaker reasoning abilities.

Moreover, some studies have further focused on addressing overthinking and underthinking in long CoTs. (Wang et al., 2025b; Sui et al., 2025) introduced thought switching penalties to influence the token decoding probability distribution early in CoT generation, reducing initial thought-switching. Similarly, (Chen et al., 2024) presented an efficiency metric to evaluate each token's contribution to accuracy and used length preference optimization to achieve more efficient CoT patterns. However, both approaches depend on auxiliary judgments from more powerful reasoning models, making the performance improvements inherently constrained by the capabilities of the reference models.

## 7 CONCLUSION

In this paper, we present AdapThink, an efficient post-training framework for reasoning models that adaptively modulates reflection preferences based on a group's outcome correctness and reasoning process characteristics. We highlight two key insights: (1) Controlling reasoning length alone does not directly determine a model's reasoning efficiency, and (2) Homogeneous reasoning patterns may also hinder effective model learning by limiting the model's exposure to diverse problem-solving strategies. Notably, with only a 2K token limit during training, AdapThink outperforms existing baselines in both reasoning accuracy and token efficiency across various mathematical reasoning benchmarks. One limitation and potential future direction lies in exploring token-level reflection-related control beyond the current response-level. We will further conduct experiments to investigate the impact of larger reasoning models.

## ETHICS STATEMENT

All authors of this study strictly adhere to the ICLR code of ethics. Our research does not involve any potential conflicts of interest or sponsorship issues. We have carefully considered and addressed concerns related to discrimination, bias, and fairness in our methodology. The study raises no privacy or security concerns, maintains full legal compliance, and upholds the highest standards of research integrity. All experimental procedures and data handling practices follow established ethical guidelines for machine learning research.

## REPRODUCIBILITY STATEMENT

To ensure full reproducibility of our results, we provide comprehensive implementation details of the proposed BAPO training algorithm in the supplementary materials. All experimental settings, hyperparameters, and dataset specifications are clearly documented. For our theoretical contributions, complete proofs and clear explanations of all assumptions are included in the appendix. Code and data will be made available upon acceptance to facilitate replication of our findings.

## THE USE OF LARGE LANGUAGE MODELS

In this research, we employed LLMs solely as language editing tools to improve the clarity and readability of our manuscript. LLMs were used for grammar checking, style refinement, and language polishing purposes only. All core research ideas, experimental design, analysis, and conclusions are entirely the original work of the authors. The use of LLMs did not contribute to the conceptual or technical content of this study.

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

# A  EXAMPLES OF BASE MODEL'S THINKING PROCESS

## A.1  EXAMPLE 1: OVERTHINKING ON A SIMPLE PROBLEM

In this example, the model (DeepSeek Distilled Qwen 1.5B) was presented with a straightforward probability problem. While it reached the correct answer early in its reasoning process (using only 479 tokens), it continued to reflect unnecessarily. The model's self-validation tendencies, marked by frequent use of phrases like "Wait", "check", and "verify", led to multiple alternative approaches and verifications. This resulted in consuming 2,613 tokens—more than five times the tokens needed for the initial correct solution.

---

**Overthinking Example**

**Question**

A bag contains 4 red balls and 3 blue balls. Two balls are drawn without replacement. What is the probability that both balls are of different colors?

**Model Response**

**Thinking Process**

Okay, let's see. Well, I can think ... **Alternatively**, maybe I should ... **Wait**, is it right? ... **Alternatively**, maybe it is easier to ... So, the probability is 4/7.
But ... **Alternatively** ... But let me **check** both approaches ... **Alternatively** ... Approach 1 ... Wait ... Let me **check** ... Approach 2 ... **Alternatively** ... **Wait**, but let me just **verify** ... So all methods lead to 4/7.
\*\*Final Answer\*\*:

The probability that both balls are of different colors is $\boxed{\dfrac{4}{7}}$.

All methods confirm the probability that both balls are of different colors is $\boxed{\dfrac{4}{7}}$.

**Correctness**

Correct

**Tokens Consumption**

2613 (479 for first generating the correct answer.)

**Reflection Words Count**

19

---

## A.2 EXAMPLE 2: UNDERTHINKING ON A COMPLEX PROBLEM

The second example shows the model attempting a more challenging modular arithmetic problem. Here, the model exhibited frequent but unproductive shifts in thinking, marked by numerous "Wait" and "Alternatively" statements. Unlike the first example where excessive reflection was merely inefficient, it reached the maximum token limit (4,096 token) before completing its calculation, resulting in an incorrect answer.

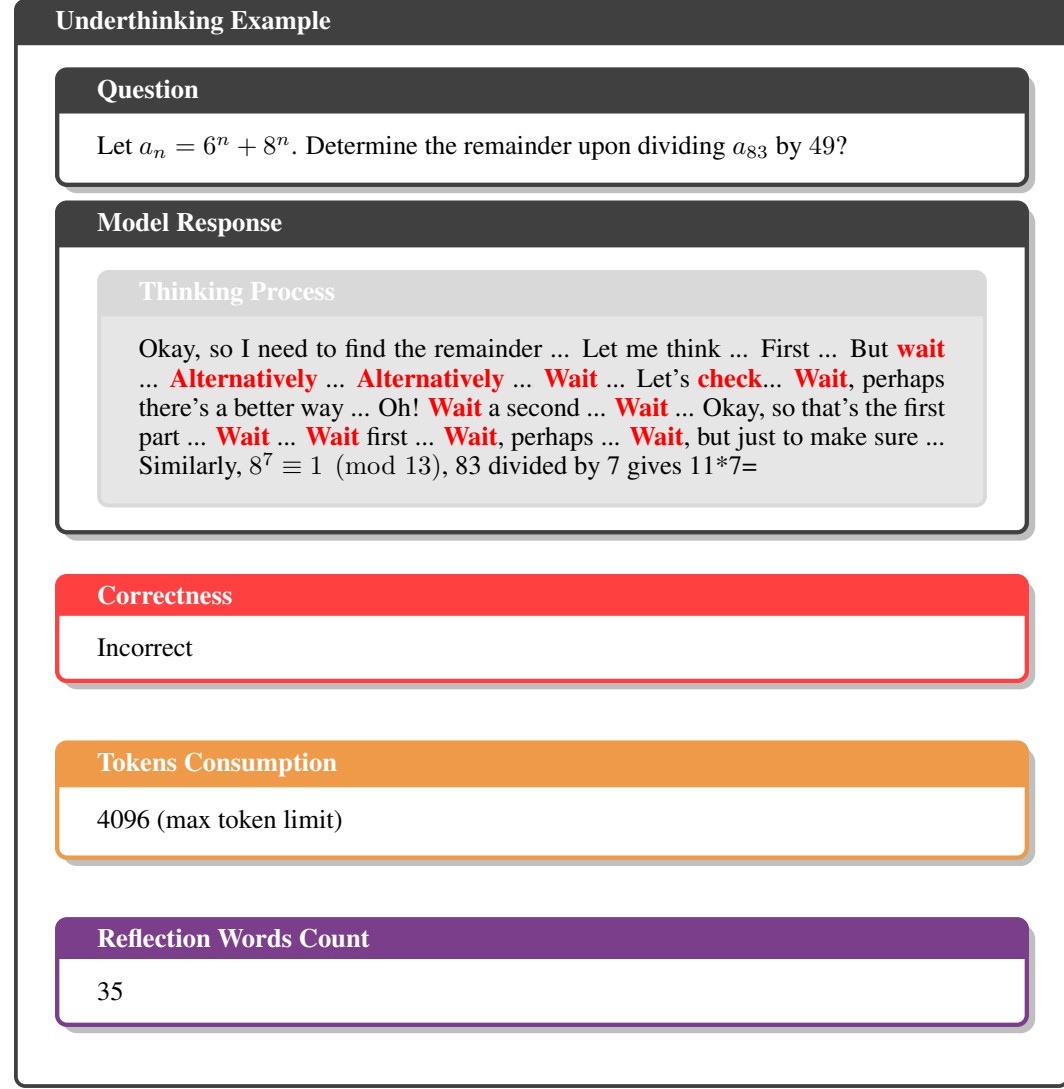

**Underthinking Example**

**Question**

Let $a_n = 6^n + 8^n$. Determine the remainder upon dividing $a_{83}$ by $49$?

**Model Response**

**Thinking Process**

Okay, so I need to find the remainder ... Let me think ... First ... But **wait** ... **Alternatively** ... **Alternatively** ... **Wait** ... Let's **check**... **Wait**, perhaps there's a better way ... Oh! **Wait** a second ... **Wait** ... Okay, so that's the first part ... **Wait** ... **Wait** first ... **Wait**, perhaps ... **Wait**, but just to make sure ... Similarly, $8^7 \equiv 1 \pmod{13}$, 83 divided by 7 gives 11*7=

**Correctness**

Incorrect

**Tokens Consumption**

4096 (max token limit)

**Reflection Words Count**

35

## B ADAPTHINK ALGORITHM DETAILS

**Pseudo Code.** We provide the complete training process of AdapThink in Algorithm 1 and the sampling strategy in Algorithm 2.

**Objective.** Following the setting of GRPO (Guo et al., 2025), the objective is formulated as:

$$\frac{1}{G} \sum_{i=1}^{G} \frac{1}{|y_i|} \sum_{t=1}^{|y_i|} \min\left(\rho_{i,t}(\theta)\hat{A}_{i,t}, \text{clip}(\rho_{i,t}(\theta), 1-\varepsilon, 1+\varepsilon)\hat{A}_{i,t}\right) - \beta \cdot \mathbb{D}_{\text{KL}}(\pi_\theta||\pi_{\text{ref}}) \quad (7)$$

where $\mathcal{G} = \{y_1, y_2, \ldots, y_G\}$ represents a $G$-size group of responses sampled from $\pi_{\theta_t}(\cdot|x)$ for each input $x$; $\rho_{i,t}(\theta)$ is the probability ratio $\frac{\pi_\theta\left(y_i^t|y_i^{<t},x\right)}{\pi_{\theta_{\text{old}}}\left(y_i^t|y_i^{<t},x\right)}$ between current policy and old policy $\pi_{\theta_{\text{old}}}$

for the $i$-th responses' $t$-th token, $\varepsilon$ limits the magnitude of policy updates; and $\mathbb{D}_{\text{KL}}$ constrains the policy $\pi_\theta$ from deviating too far from a reference policy $\pi_{\text{ref}}$. Crucially, $\hat{A}_{i,t}$ denotes the estimated advantage of response $y$ for input $x$ within group $\mathcal{G}$, which is derived from the standardization of rewards using the statistical properties of group $\mathcal{G}$. For the $i$-th response $y_i \in \mathcal{G}$ with reward $r_i = r(x, y_i)$, the estimated advantage is:

$$\hat{A}_{i,t} = \frac{r_i - \text{mean}(\{r_\ell\})}{\sqrt{\text{std}^2(\{r_\ell\}) + \varepsilon}} \tag{8}$$

where $\text{mean}(\{r_\ell\})$ and $\text{std}^2(\{r_\ell\})$ are the empirical mean and variance of rewards in group $\mathcal{G}$, respectively.

For the reward $r_i$, we incorporate two components: the outcome reward $r_{\text{outcome}}$ and the reflection-related process reward $r_{\text{ref}}$. The definition of $r_{\text{ref}}$ is provided in Equation 4. The reward $r_{\text{outcome}}$ is defined as follows:

$$r_{\text{outcome}}(x, y) = \begin{cases} 1, & \text{if } y \text{ is correct} \\ 0, & \text{otherwise} \end{cases}.$$

---

**Algorithm 1** AdapThink Training Framework

---

**Require:** Pre-trained model $\pi_\theta$, Dataset $\mathcal{D}$, Upsample factor $K$
**Ensure:** Updated model $\pi_\theta$
1: **for** each $(x, y^*) \in \mathcal{D}$ **do**
2:    $\mathcal{G} \leftarrow \text{DIVERSITYSAMPLING}(x, K)$ {Using Algorithm 2}
3:    $\varphi \leftarrow \frac{1}{|\mathcal{G}|} \sum_{i=1}^{|\mathcal{G}|} \mathbb{I}(\hat{y}_i = y^*)$ {Group Correctness}
4:    $\mathcal{G}_\mathcal{T} \leftarrow \{\hat{y}_i \in \mathcal{G} : \hat{y}_i = y^*\}, \mathcal{G}_\mathcal{F} \leftarrow \{\hat{y}_i \in \mathcal{G} : \hat{y}_i \neq y^*\}$
5:    **for** $\mathcal{G}_{\text{sub}} \in \{\mathcal{G}_\mathcal{T}, \mathcal{G}_\mathcal{F}\}$ **do**
6:       $\mu_l(\mathcal{G}_{\text{sub}}) \leftarrow \frac{1}{|\mathcal{G}_{\text{sub}}|} \sum_{\hat{y}_i \in \mathcal{G}_{\text{sub}}} l(\hat{y}_i)$ {Group Length Baseline}
7:       $\mu_r(\mathcal{G}_{\text{sub}}) \leftarrow \frac{1}{|\mathcal{G}_{\text{sub}}|} \sum_{\hat{y}_i \in \mathcal{G}_{\text{sub}}} r(\hat{y}_i)$ {Group Reflection Words Baseline}
8:    **end for**
9:    **for** each $\hat{y}_i \in \mathcal{G}$ **do**
10:     $r_{\text{ref}}(\hat{y}_i) \leftarrow \omega(\varphi) \cdot \left[ \frac{\mu_l(\mathcal{G}(\hat{y}_i)) - l(\hat{y}_i)}{\mu_l(\mathcal{G}(\hat{y}_i))} + \frac{\mu_r(\mathcal{G}(\hat{y}_i)) - r(\hat{y}_i)}{\mu_r(\mathcal{G}(\hat{y}_i))} \right]$ {Reasoning Process Reward}
11:     $r_{\text{total}}(\hat{y}_i) \leftarrow r_{\text{answer}}(\hat{y}_i) + r_{\text{ref}}(\hat{y}_i)$
12:    **end for**
13:    $\pi_\theta \leftarrow \text{GRPO}(\pi_\theta, \mathcal{G}, \{r_{\text{total}}(\hat{y}_i)\})$ (Equation 7)
14: **end for**

---

## C   More Results

**Analysis of Detailed Reflection Words Distribution.** To provide deeper insights into *how different post-training methods affect the model's reasoning behavior*, we analyze the distribution of reflection words across correct and incorrect responses, as shown in Figure 6. For most reflection-related words, we observe that AdapThink significantly reduces the frequency in both correct and incorrect responses compared to the base model, suggesting a more efficient reasoning process. In contrast, GRPO and TLB exhibit less pronounced changes in these reflection words; LCPO shows elevated counts of pause words, particularly in incorrect responses.

**Analysis of More Ablation.** We evaluate AdapThink on two out-of-distribution (OOD) tasks, as shown in Table 5. The results demonstrate that models fine-tuned on the MATH dataset achieve improved PASS@1 performance on other OOD tasks compared to the baseline, while significantly reducing token consumption. Additionally, we track the accuracy and token length of three Adap-Think variants on the AIME2025 benchmark across checkpoints at 50-step intervals. Our findings reveal: (1) AdapThink (No weight) exhibits consistent token reduction behavior, leading to performance gains in the first 250 steps but causing significant accuracy degradation after 250 steps. (2) AdapThink (Sequential) shows that controlling non-reflective vocabulary has minimal impact on the results.

---

**Algorithm 2** Diversity-Aware Sampling Strategy

---

**Require:** Question $x$, Upsample factor $K$, Target size $|\mathcal{G}|$
**Ensure:** Diverse sample set $\mathcal{G}$
 1: Stage 1: Upsampling
 2: $\mathcal{G}' \leftarrow$ Generate $K \times |\mathcal{G}|$ responses from $\pi_\theta(\cdot|x)$
 3: $\mathcal{S} \leftarrow$ Partition $(l, r)$ space into 4 bins
 4: Stage 2: Downsampling
 5: $\mathcal{G}_{\text{best}} \leftarrow \emptyset$, $\mathcal{H}_{\text{max}} \leftarrow 0$
 6: **if** $\varphi(\mathcal{G}') \in (0, 1)$ **then**
 7:     **for** each subset $\mathcal{G}_{\text{cand}} \subseteq \mathcal{G}'$ with $|\mathcal{G}_{\text{cand}}| = |\mathcal{G}|$ **do**
 8:         **if** $\varphi(\mathcal{G}_{\text{cand}}) \in (0, 1)$ **and** $\mathcal{H}(\mathcal{G}_{\text{cand}}) > \mathcal{H}_{\text{max}}$ **then**
 9:             $\mathcal{G}_{\text{best}} \leftarrow \mathcal{G}_{\text{cand}}$, $\mathcal{H}_{\text{max}} \leftarrow \mathcal{H}(\mathcal{G}_{\text{cand}})$
10:         **end if**
11:     **end for**
12: **else**
13:     **for** each subset $\mathcal{G}_{\text{cand}} \subseteq \mathcal{G}'$ with $|\mathcal{G}_{\text{cand}}| = |\mathcal{G}|$ **do**
14:         **if** $\mathcal{H}(\mathcal{G}_{\text{cand}}) > \mathcal{H}_{\text{max}}$ **then**
15:             $\mathcal{G}_{\text{best}} \leftarrow \mathcal{G}_{\text{cand}}$, $\mathcal{H}_{\text{max}} \leftarrow \mathcal{H}(\mathcal{G}_{\text{cand}})$
16:         **end if**
17:     **end for**
18: **end if**
19: **return** $\mathcal{G} \leftarrow \mathcal{G}_{\text{best}}$

---

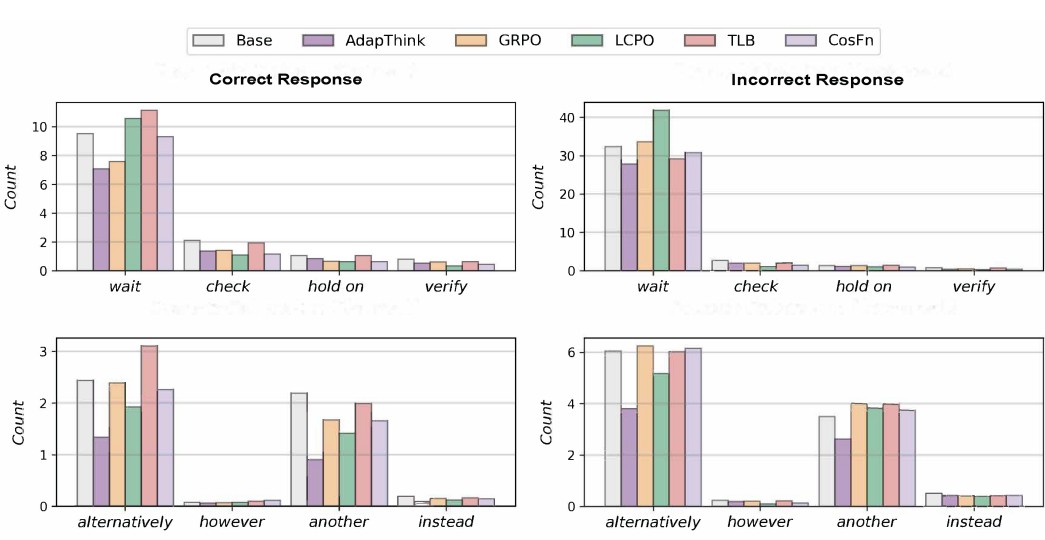

Figure 6: Distribution of reflection words across different post-training methods for correct and incorrect responses in MATH-500 datasets.

Table 4: **Ablation studies of Diversity-aware Sampling**.

| Variants | Accuracy |
|----------|----------|
| LCPO | 27.6 |
| LCPO+$\mathcal{H}$ | 28.4 |
| DAST | 29.8 |
| DAST+$\mathcal{H}$ | 32.8 |

Table 5: **Performance Comparison on Out-of-Distribution (OOD) Benchmarks** for AdapThink.

| Model | LSAT | | GPQA | |
|-------|--------|-----------|--------|-----------|
| | Pass@1 | Avg Token | Pass@1 | Avg Token |
| Deepseek 1.5B | 0.2462 | 7481 | **0.2942** | 6052 |
| +AdapThink | **0.2478** | **6810** | 0.2830 | **4436** |
| Deepseek 7B | 0.4168 | 6723 | 0.4561 | 5759 |
| +AdapThink | **0.4283** | **5982** | **0.4838** | **4944** |

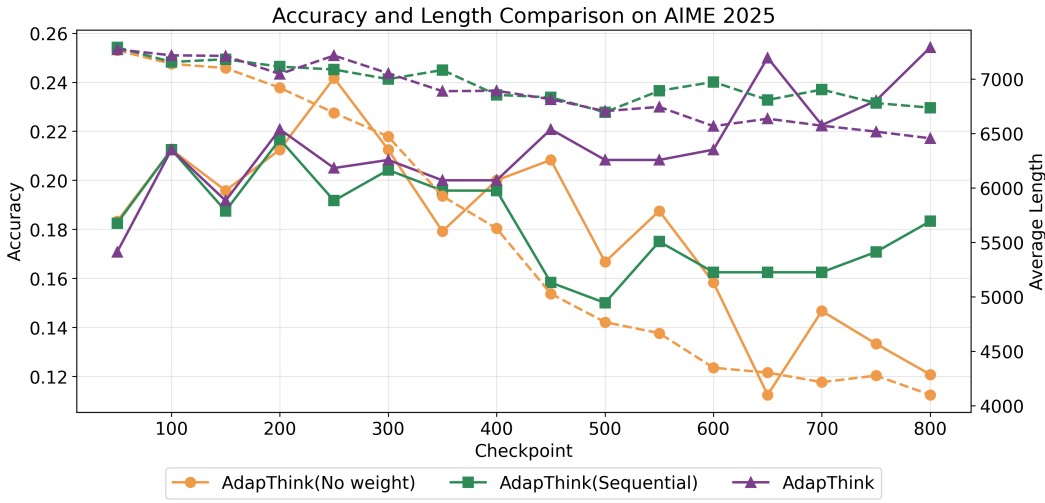

Figure 7: Evaluation comparison among three AdapThink variants across checkpoints at 50-step intervals.

## D  HYPERPARAMETER CONFIGURATION

The core hyperparameters used in AdapThink are summarized in Table 6. The training is on 8 NVIDIA H100 GPUs, taking approximately 22 hours to complete 1000 steps for 1.5B model and 1 day 16 hours for 7B model.

Table 6: Core hyperparameters for AdapThink post-training configuration

| Category | Parameter | Value |
|---|---|---|
| Model Settings | Base Model | DeepSeek-R1-Distill-Qwen-1.5B |
| | Torch Dtype | bfloat16 |
| | Max Sequence Length | 1024 |
| | Max New Tokens | 2048 |
| Training Settings | Learning Rate | 2e-6 |
| | Number of Epochs | 5 |
| | Batch Size | 8 |
| | Gradient Accumulation Steps | 8 |
| | Warmup Steps | 20 |
| LoRA Settings | LoRA Rank (r) | 32 |
| | LoRA Alpha | 32 |
| RL Settings | Algorithm | GRPO |
| | KL Coefficient | 0.15 |
| | Number of Generations | 12 |
| | Target Generations | 8 |
| Generation Settings | Temperature | 0.7 |
| | Top-p | 0.95 |

# E    EXAMPLE OF BASE MODEL AND ADAPTHINK

---

**DeepSeek-distilled Qwen 1.5B Example (from AIME 2025)**

**Question**

Find the number of ordered pairs $(x, y)$, where both $x$ and $y$ are integers between $-100$ and $100$, inclusive, such that $12x^2 - xy - 6y^2 = 0$.

**Model Response**

**Thinking Process**

Alright, so I have this problem here: ... Hmm, okay. Let me try to figure this out step by step.

First, I notice that this is a quadratic equation in terms of x and y. Maybe I can factor it or rewrite it .... Let me write it down again:

$$12x^2 - xy - 6y^2 = 0$$

**Alternatively**, maybe factor the quadratic expression as $(ax + by)(cx + dy) = 0$. Let me try that...
**Alternatively**, perhaps I can write the equation as: $12x^2 - xy - 6y^2 = 0$...
**Alternatively**, maybe factor it as $(ax + by)(cx + dy) = 0$...
**Alternatively**, perhaps factor out a 3:

$$3(4x^2 - \frac{1}{3}xy - 2y^2) = 0$$

But that doesn't seem helpful.
**Alternatively**, maybe factor as $(4x + ay)(3x + by) = 0$ ...
So, I think I can confidently say the answer is 117. **\*\*Final Answer\*\***: $\boxed{117}$.

We need to find the number of ordered pairs $(x, y)$ where both $x$ and $y$ are integers between $-100$ and $100$, inclusive, such that $12x^2 - xy - 6y^2 = 0$ ... The total number of solutions is:

$$67 + 51 - 1 = 117$$

Thus, the number of ordered pairs $(x, y)$ is: $\boxed{117}$,

**Correctness**

Correct

**Tokens Consumption**

**7377**

**Reflection Words Count**

**46**

---

**AdapThink Example (from AIME 2025)**

### Question

Find the number of ordered pairs $(x, y)$, where both $x$ and $y$ are integers between $-100$ and $100$, inclusive, such that $12x^2 - xy - 6y^2 = 0$.

### Model Response

**Thinking Process**

Okay, so I need to find the number of ordered pairs $(x, y)$ where both $x$ and $y$ are integers between -100 and 100, inclusive. The condition given is $12x^2 - xy - 6y^2 = 0$. Hmm, let me try to figure this out step by step.

First, looking at the equation: $12x^2 - xy - 6y^2 = 0$. It seems like a quadratic in terms of x and y. Maybe I can factor it or rearrange it somehow. ... Let's see:

$$12x^2 - xy - 6y^2 = 0$$

Divide both sides by 3:

$$4x^2 - \frac{xy}{3} - 2y^2 = 0$$

**Wait**, that doesn't help much because we have a fraction. Maybe factoring differently. ...

So, seems consistent. Therefore, the number of ordered pairs is 117. **Final Answer**: The number of ordered pairs is $\boxed{117}$.

To find the number of ordered pairs $(x, y)$ where both $x$ and $y$ are integers between -100 and 100, inclusive, such that $12x^2 - xy - 6y^2 = 0$, we start by ... Thus, we subtract 1 to avoid double-counting. The total number of ordered pairs is:

$$51 + 67 - 1 = 117$$

Thus, the number of ordered pairs is $\boxed{117}$.

### Correctness

Correct

### Tokens Consumption

**4502**

### Reflection Words Count

**19**

