# OpenReview forum: "Beyond Fixed Budgets: Dynamic Reasoning Efficiency Reward for Large Language Model"
_ICLR.cc/2026/Conference — ICLR 2026 Conference Withdrawn Submission_

### Official Review · Reviewer_9FJf · 2025-10-29

**Soundness:** 3
**Presentation:** 2
**Contribution:** 2
**Rating:** 2
**Confidence:** 5

**Summary:**

This paper proposes AdapThink, a post-training framework designed to enhance reasoning efficiency in large language models (LLMs) by adaptively balancing “slow thinking” and token efficiency. Unlike prior works that apply static token budgets or uniform length penalties, AdapThink introduces:
1. Group-relative reasoning process rewards, which quantify reasoning efficiency using the statistical distribution of reasoning length and reflection-related words.
2. Diversity-aware sampling, which promotes diverse reasoning trajectories while preserving accuracy.
Experiments on multiple mathematical reasoning datasets (e.g., MATH-500, AIME 2024/2025, AMC 2023) show that AdapThink outperforms GRPO and other length-control methods, achieving 6–7% higher accuracy with fewer tokens.

**Strengths:**

Clear motivation: The paper addresses a timely challenge in reasoning efficiency for LLMs, especially as “slow thinking” paradigms (e.g., O1, DeepSeek R1) become mainstream.

The proposed method is technically sound: The group-relative reasoning reward is reasonable, which shifts from token-level control to process-level adaptation.

The experiments show that the proposed AdapThink consistently improves performance on both reasoning accuracy and token efficiency across datasets.

**Weaknesses:**

1. The limited novelty. While AdapThink introduces new reward formulations, the overall framework (RL-based post-training with process rewards) resembles prior GRPO-like or MERA approaches, with only moderate innovation in principle. It is some tricks more than a solid method.

2. The adopted models are small-scale (1.5B and 7B). However, the overthinking problems mainly happened with large-scale models with deep reasoning abilities.

3. Experiments are limited to mathematical reasoning. Although an OOD test is included, these evaluations are shallow and do not convincingly demonstrate generalization to non-mathematical reasoning.

**Questions:**

1. Does AdapThink maintain stability when scaling to much larger context lengths?
2. How sensitive is AdapThink to the choice of the reflection-word list or the threshold?

---

### Official Review · Reviewer_6DnZ · 2025-10-30

**Soundness:** 2
**Presentation:** 3
**Contribution:** 2
**Rating:** 2
**Confidence:** 4

**Summary:**

The paper tackles overthinking/underthinking in LLM reasoning and proposes AdapThink, a post-training framework that adaptively regulates “slow thinking” without fixed length budgets. It combines a group-relative process reward (using response length and reflection-word frequency) with a diversity-aware sampling scheme.

**Strengths:**

1. The authors conducted systematic and comprehensive ablation experiments for the proposed method.
2. The paper is weel-written and orignized.

**Weaknesses:**

1.	The “group-relative process reasoning
reward” method proposed in this paper, judging from its formulation (Eq. 4), appears to primarily guide the model to optimize its reasoning process on easy problems (i.e., those the model can already answer correctly), while providing little effective guidance for the reasoning on difficult problems.  I did not see results demonstrating that training with this method effectively alters the distribution pattern presented in Figure 2.
2.	Comparing the official DeepSeek-R1 results (https://huggingface.co/deepseek-ai/DeepSeek-R1-Distill-Qwen-7B) with the data in Table 1 of this paper, the “Original” model after GRPO training actually underperforms the pre-training base model on certain benchmarks. For example, DeepSeek-R1-Distill-Qwen-7B attains 92.8 on MATH-500 (pass@1) and 55.5 on AIME 2024 (pass@1), whereas the “Original” model in this paper achieves only 86.9 and 53.1, respectively. This result is somewhat anomalous and may affect the validity of the comparative experiments reported.
3.	From Fig. 4, the length of the model’s responses shows a steadily decreasing trend. However, according to the DeepSeek-R1 report and related replication studies, performance improvements during training are typically accompanied by increasing response length. Therefore, if the experiment in Fig. 4 were continued, would the response length exhibit an inflection point?

**Questions:**

See weakness.

---

### Official Review · Reviewer_udEf · 2025-10-31

**Soundness:** 2
**Presentation:** 3
**Contribution:** 2
**Rating:** 4
**Confidence:** 3

**Summary:**

This paper propose AdapThink framework to fix overthinking/underthinking in LLMs. It use group-level reasoning process reward that compare each response with group average length and reflection words. Also use diversity sampling to get varied reasoning patterns. Test on DeepSeek-Qwen models, train with 2K tokens and get 27% improvement over GRPO baseline.

**Strengths:**

- This paper is well-written and proposes a simple and intuitive method that is easy to understand and follow.
- This paper conducts comprehensive experiments and the ablation studies have shown that AdapThink can teach the model general principles, not just compress output.

**Weaknesses:**

- This method is too simplistic - just comparing with group average and counting reflection words. No fundamental improvement over existing approaches.
- This method was only tested on two models from the same family (DeepSeek-Qwen 1.5B/7B). Meanwhile, the improvements are marginal, which is not sufficient to justify the effectiveness of AdapThink.

**Questions:**

Have you considered testing AdapThink on non-math domains like code generation, commonsense reasoning？

---

### Official Review · Reviewer_xE1y · 2025-11-03

**Soundness:** 2
**Presentation:** 3
**Contribution:** 3
**Rating:** 6
**Confidence:** 3

**Summary:**

This paper proposes a post‑training framework for reasoning LLMs that augments GRPO with a group‑relative process reward that favors responses whose length and count of reflection words are below their group mean, with a cosine weight that increases when group accuracy is high. They also propose a two‑stage, diversity‑aware sampling procedure that upsamples responses, then down‑samples to maximize entropy over binned pairs while keeping group correctness non‑degenerate.

**Strengths:**

Instead of imposing static budgets, the method uses the group’s length and reflection word distributions and the model’s confidence to determine how much slow thinking is appropriate. The proposed method achieves a 27 % improvement in convergence reward over the GRPO baseline for 2k-token context and 12.6 % accuracy improvement over the base model for 32k-token context. On AIME‑2025 and MATH‑500, AdapThink outperforms baselines such as GRPO and LCPO both in pass@1 and token consumption. These results indicate that the method does not merely cut tokens, but translates the efficiency into measurable accuracy gains.

**Weaknesses:**

Using counts of hand‑picked words as the target for reflection control is brittle, language‑ and style‑dependent, and can easily be gamed. The ablation itself shows penalizing check/verify hurts, which contradicts the claim that reflection words are a good proxy to control.

The observation that incorrect answers have more tokens and reflection markers does not justify a reward that penalizes those markers causal‑mechanistically. Competing works already handle difficulty and underthinking more directly.

**Questions:**

Please provide a theoretical argument why group‑relative proxies of style (length, word‑counts) should generalize across tasks/model scales, or replace them with learned process rewards.

Please elaborate more on the novelty with respect to related work such as LCPO, DAST, DEER, MERA, TIP/NoWait, CoD/C3oT.

---

### Note · Authors · 2025-12-04

I have read and agree with the venue's withdrawal policy on behalf of myself and my co-authors.